# A Bilingual Generative Transformer for Semantic Sentence Embedding

## Abstract

Semantic sentence embedding models encode natural language sentences into vectors, such that closeness in embedding space indicates closeness in the semantics between the sentences. Bilingual data offers a useful signal for learning such embeddings: properties shared by both sentences in a translation pair are likely semantic, while divergent properties are likely stylistic or language-specific. We propose a deep latent variable model that attempts to perform source separation on parallel sentences, isolating what they have in common in a latent semantic vector, and explaining what is left over with language-specific latent vectors. Our proposed approach differs from past work on semantic sentence encoding in two ways. First, by using a variational probabilistic framework, we introduce priors that encourage source separation, and can use our model's posterior to predict sentence embeddings for monolingual data at test time. Second, we use high-capacity transformers as both data generating distributions and inference networks – contrasting with most past work on sentence embeddings. In experiments, our approach substantially outperforms the state-of-the-art on a standard suite of unsupervised semantic similarity evaluations. Further, we demonstrate that our approach yields the largest gains on more difficult subsets of these evaluations where simple word overlap is not a good indicator of similarity.

## 1 Introduction

Learning useful representations of language has been a source of recent success in natural language processing (NLP). Much work has been done on learning representations for words (Mikolov et al., 2013; Pennington et al., 2014) and sentences (Kiros et al., 2015; Conneau et al., 2017). More recently, deep neural architectures have been used to learn contextualized word embeddings (Peters et al., 2018; Devlin et al., 2018) which have enabled state-of-the-art results on many tasks. We focus on learning semantic *sentence* embeddings in this paper, which play an important role in many downstream applications. Since they do not require any labelled data for fine-tuning, sentence embeddings are useful for a variety of problems right out of the box. These include Semantic Textual Similarity (STS; Agirre et al. (2012)), mining bitext (Zweigenbaum et al., 2018), and paraphrase identification (Dolan et al., 2004). Semantic similarity measures also have downstream uses such as fine-tuning machine translation systems (Wieting et al., 2019a).

There are three main ingredients when designing a sentence embedding model: the architecture, the training data, and the objective function. Many architectures including LSTMs (Hill et al., 2016; Conneau et al., 2017; Schwenk & Douze, 2017; Subramanian et al., 2018), Transformers (Cer et al., 2018; Reimers & Gurevych, 2019), and averaging models (Wieting et al., 2016a; Arora et al., 2017) have found success for learning sentence embeddings. The choice of training data and objective are intimately intertwined, and there are a wide variety of options including next-sentence prediction (Kiros et al., 2015), machine translation (Espana-Bonet et al., 2017; Schwenk & Douze, 2017; Schwenk, 2018; Artetxe & Schwenk, 2018), natural language inference (NLI) (Conneau et al., 2017), and multi-task objectives which include some of the previously mentioned objectives (Cer et al., 2018) as well as additional tasks like constituency parsing (Subramanian et al., 2018).

Surprisingly, despite ample testing of more powerful architectures, the best performing models for many sentence embedding tasks related to semantic similarity often use simple architectures that are mostly agnostic to the interactions between words. For instance, some of the top performing

techniques use word embedding averaging (Wieting et al., 2016a), character n-grams (Wieting et al., 2016b), and subword embedding averaging (Wieting et al., 2019b) to create representations. These simple approaches are competitive with much more complicated architectures on in-domain data and generalize well to unseen domains, but are fundamentally limited by their inability to capture word order. Training these approaches generally relies on discriminative objectives defined on paraphrase data (Ganitkevitch et al., 2013; Wieting & Gimpel, 2018) or bilingual data (Wieting et al., 2019b). The inclusion of latent variables in these models has also been explored (Chen et al., 2019).

Intuitively, bilingual data in particular is promising because it potentially offers a useful signal for learning the underlying semantics of sentences. Within a translation pair, properties shared by both sentences are more likely semantic, while those that are divergent are more likely stylistic or language-specific. While previous work learning from bilingual data perhaps takes advantage of this fact *implicitly*, the focus of this paper is modelling this intuition *explicitly*, and to the best of our knowledge, this has not not been explored in prior work. Specifically, we propose a deep generative model that is encouraged to perform *source separation* on parallel sentences, isolating what they have in common in a latent *semantic embedding* and explaining what is left over with *language-specific latent vectors*. At test time, we use inference networks (Kingma & Welling, 2013) for approximating the model's posterior on the semantic and source-separated latent variables to encode monolingual sentences. Finally, since our model and training objective are generative, our approach does not require knowledge of the distance metrics to be used during evaluation,[1] and it has the additional property of being able to generate text.

In experiments, we evaluate our probabilistic source-separation approach on a standard suite of STS evaluations. We demonstrate that the proposed approach is effective, most notably allowing the learning of high-capacity deep transformer architectures (Vaswani et al., 2017) while still generalizing to new domains, significantly outperforming a variety of state-of-the-art baselines . Further, we conduct a thorough analysis by identifying subsets of the STS evaluation where simple word overlap is not able to accurately assess semantic similarity. On these most difficult instances, we find that our approach yields the largest gains, indicating that our system is modeling interactions between words to good effect. We also find that our model better handles cross-lingual semantic similarity than multilingual translation baseline approaches, indicating that stripping away language-specific information allows for better comparisons between sentences from different languages.

Finally, we analyze our model to uncover what information was captured by the source separation into the semantic and language-specific variables and the relationship between this encoded information and language distance to English. We find that the language-specific variables tend to explain more superficial or language-specific properties such as overall sentence length, amount and location of punctuation, and the gender of articles (if gender is present in the language), but semantic and syntactic information is more concentrated in the shared semantic variables, matching our intuition. Language distance has an effect as well, where languages that share common structures with English put more information into the semantic variables, while more distant languages put more information into the language-specific variables. Lastly, we show outputs generated from our model that exhibit its ability to do a type of *style transfer*.

## 2 MODEL

Our proposed training objective leverages a generative model of parallel text in two languages (e.g. English (en) and French (fr)) that form a pair consisting of an English sentence $x_{en}$ and a French sentence $x_{fr}$. Importantly, this generative process utilizes three underlying latent vectors: language-specific variation variables (language variables) $z_{fr}$ and $z_{en}$ respectively for each side of the translation, as well as a shared semantic variation variable (semantic variable) $z_{sem}$. In this section we will first describe the generative model for the text and latent variables. In the following section we will describe the inference procedure of $z_{sem}$ given an input sentence, which corresponds to our core task of obtaining sentence embeddings useful for downstream tasks such as semantic similarity.

Further, by encouraging the model to perform this source separation, the learned semantic encoders will more crisply represent the underlying semantics, increasing performance on downstream semantic tasks.

---

[1] In other words, we don't assume cosine similarity as a metric, though it does work well in our experiments.

The generative process of our model, the Bilingual Generative Transformer (BGT), is depicted in Figure 1 and its computation graph is shown in Figure 2. First, we sample latent variables $\langle z_{fr}, z_{en}, z_{sem}\rangle$, where $z_i \in \mathbb{R}^k$, from a multivariate Gaussian prior $\mathcal{N}(0, I_k)$. These variables are then fed into a decoder that samples sentences; $x_{en}$ is sampled conditioned on $z_{sem}$ and $z_{en}$, while $x_{fr}$ is sampled conditioned on $z_{sem}$ and $z_{fr}$. Because sentences in both languages will use $z_{sem}$ in generation, we expect that in a well-trained model this variable will encode semantic, syntactic, or stylistic information shared across both sentences, while $z_{fr}$ and $z_{en}$ will handle any language-specific peculiarities or specific stylistic decisions that are less central to the sentence meaning and thus do not translate across sentences. In the following section, we further discuss how this is explicitly encouraged by the learning process.

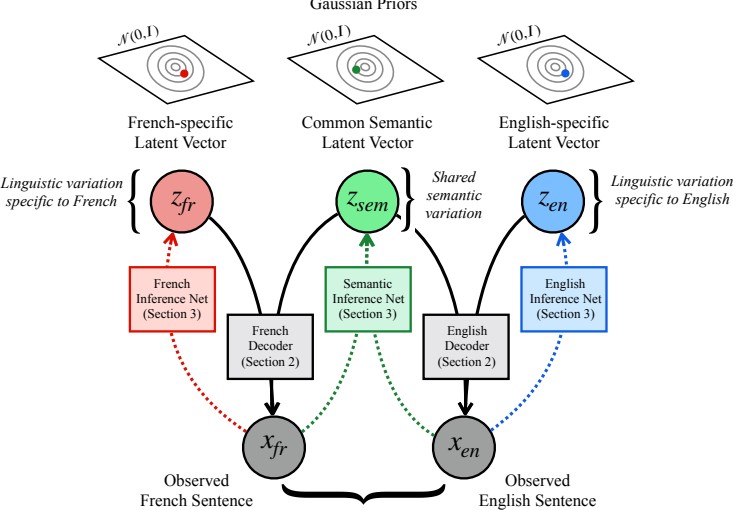

Figure 1: The generative process of our model. Latent variables modeling the linguistic variation in French and English, $z_{fr}$ and $z_{en}$, as well as a latent variable modeling the common semantics, $z_{sem}$, are drawn from a multivariate Gaussian prior. The observed text in each language is then conditioned on its language-specific variable and $z_{sem}$.

**Decoder Architecture.**  Many latent variable models for text use LSTMs (Hochreiter & Schmidhuber, 1997) as their decoders (Yang et al., 2017; Ziegler & Rush, 2019; Ma et al., 2019). However, state-of-the-art models in neural machine translation have seen increased performance and speed using deep Transformer architectures. We also found in our experiments (see Appendix C for details) that Transformers led to increased performance in our setting, so they are used in our main model.

We use two decoders in our model, one for modelling $p(x_{fr}|z_{sem}, z_{fr}; \theta)$ and one for modeling $p(x_{en}|z_{sem}, z_{en}; \theta)$. These decoders are depicted on the right side of Figure 2. Each decoder takes in two latent variables, a language variable and a semantic variable. These variables are concatenated together prior to being used by the decoder for reconstruction. We explore four ways of using this latent vector: (1) Concatenate it to the word embeddings (Word) (2) Use it as the initial hidden state (Hidden, LSTM only) (3) Use it as you would the *attention context vector* in the traditional sequence-to-sequence framework (Attention) and (4) Concatenate it to the hidden state immediately prior to computing the logits (Logit). Unlike Attention, there is no additional feedforward layer in this setting. We experimented with these four approaches, as well as combinations thereof, and report this analysis in Appendix A. From these experiments, we see that the closer the sentence embedding is to the softmax, the better the performance on downstream tasks evaluating its semantic content. We hypothesise that this is due to better gradient propagation because the sentence embedding is now closer to the error signal. Since Attention and Logit performed best, we use these in our Transformer experiments.

## 3 Learning and Inference

Our model is trained on a training set $X$ of parallel text consisting of $N$ examples, $X = \{\langle x_{en}^1, x_{fr}^1\rangle, \ldots, \langle x_{en}^N, x_{fr}^N\rangle\}$, and $Z$ is our collection of latent variables $Z = (\langle z_{en}^1, z_{fr}^1, z_{sem}^1\rangle, \ldots, \langle z_{en}^N, z_{fr}^N, z_{sem}^N\rangle)$. We wish to maximize the likelihood of the parameters of the two decoders $\theta$ with respect to the observed $X$, marginalizing over the latent variables $Z$.

$$p(X; \theta) = \int_Z p(X, Z; \theta) dZ$$

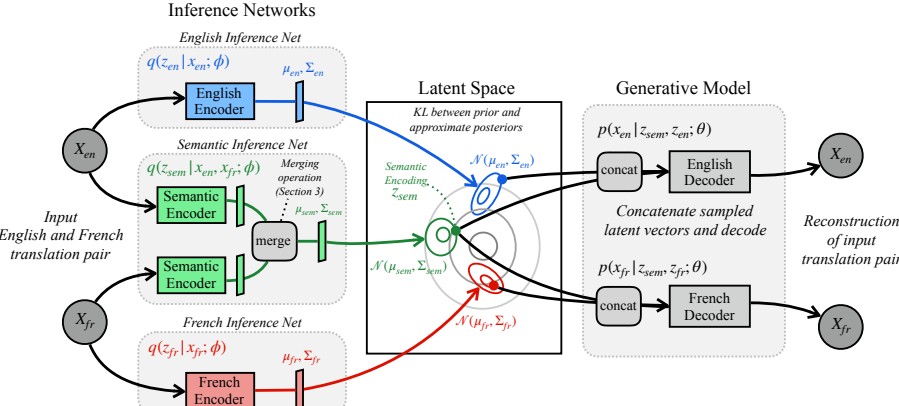

Figure 2: The computation graph for the variational lower bound used during training. The English and French text are fed into their respective inference networks and the semantic inference network to ultimately produce the language variables $z_{fr}$ and $z_{en}$ and semantic variable $z_{sem}$. Each language-specific variable is then concatenated to $z_{sem}$ and used by the decoder to reconstruct the input sentence pair.

Unfortunately, this integral is intractable due to the complex relationship between $X$ and $Z$. However, related latent variable models like variational autoencoders (VAEs (Kingma & Welling, 2013)) learn by optimizing a variational lower bound on the log marginal likelihood. This surrogate objective is called the evidence lower bound (ELBO) and introduces a variational approximation, $q$ to the true posterior of the model $p$. The $q$ distribution is parameterized by a neural network with parameters $\phi$. ELBO can be written for our model as follows:

$$\text{ELBO} = \mathbb{E}_{q(Z|X;\phi)}[\log p(X|Z;\theta)] - \text{KL}(q(Z|X;\phi)||p(Z;\theta))$$

This lower bound on the marginal can be optimized by gradient ascent by using the reparameterization trick (Kingma & Welling, 2013). This trick allows for the expectation under $q$ to be approximated through sampling in a way that preserves backpropagation.

We make several independence assumptions for $q(z_{sem}, z_{en}, z_{fr}|x_{en}, x_{fr}; \phi)$. Specifically, to match our goal of source separation, we factor $q$ as $q(z_{sem}, z_{en}, z_{fr}|x_{en}, x_{fr}; \phi) = q(z_{sem}|x_{en}, x_{fr}; \phi)q(z_{en}|x_{en})q(z_{fr}|x_{fr}; \phi)$, with $\phi$ being the parameters of the encoders that make up the inference networks, defined in the next paragraph.

Lastly, we note that the KL term in our ELBO equation encourages explaining variation that is shared by translations with the shared semantic variable and explaining language-specific variation with the corresponding language-specific variables. Information shared by the two sentences will result in a lower KL loss if it is encoded in the shared variable, otherwise that information will be replicated and the overall cost of encoding will increase.

**Encoder Architecture.** We use three inference networks as shown on the left side of Figure 2: an English inference network to produce the English language variable, a French inference network to produce the French language variable, and a semantic inference network to produce the semantic variable. Just as in the decoder architecture, we use a Transformer for the encoders.

The semantic inference network is a bilingual encoder that encodes each language. For each translation pair, we alternate which of the two parallel sentences is fed into the semantic encoder within a batch. Since the semantic encoder is meant to capture language agnostic semantic information, its outputs for a translation pair should be similar regardless of the language of the input sentence. We note that other operations are possible for combining the views each parallel sentence offers. For instance, we could feed both sentences into the semantic encoder and pool their representations. However, in practice we find that alternating works well and leave further study of this to future work.

## 4 EXPERIMENTS

### 4.1 BASELINE MODELS

We experiment with fourteen baseline models, covering both the most effective approaches for learning sentence embeddings from the literature and ablations of our own BGT model. These baselines can be split into three groups as detailed below.

**Models from the Literature (Trained on Different Data)**    We compare to well known sentence embedding models Infersent (Conneau et al., 2017), GenSen (Subramanian et al., 2018), the Universal Sentence Encoder (USE) (Cer et al., 2018), as well as BERT (Devlin et al., 2018).[2] We used the pretrained BERT model in two ways to create a sentence embedding. The first way is to concatenate the hidden states for the CLS token in the last four layers. The second way is to concatenate the hidden states of all word tokens in the last four layers and mean pool these representations. Both methods result in a 4096 dimension embedding. Finally, we compare to the newly released model, Sentence-Bert (Reimers & Gurevych, 2019). This model is similar to Infersent (Conneau et al., 2017) in that it is trained on natural language inference data, SNLI (Bowman et al., 2015). However, instead of using pretrained word embeddings, they fine-tune BERT in a way to induce sentence embeddings.[3]

**Models from the Literature (Trained on Our Data)**    These models are amenable to being trained in the exact same setting as our own models as they only require parallel text. These include the sentence piece averaging model, SP, from (Wieting et al., 2019b), which is among the best of the averaging models (i.e. compared to averaging only words or character $n$-grams) as well the LSTM model, BILSTM, from (Wieting & Gimpel, 2017). These models use a contrastive loss with a margin. Following their settings, we fix the margin to 0.4 and tune the number of batches to pool for selecting negative examples from $\{40, 60, 80, 100\}$. For both models, we set the dimension of the embeddings to 1024. For BILSTM, we train a single layer bidirectional LSTM with hidden states of 512 dimensions. To create the sentence embedding, the forward and backward hidden states are concatenated and mean-pooled. Following (Wieting & Gimpel, 2017), we shuffle the inputs with probability $p$, tuning $p$ from $\{0.3, 0.5\}$.

We also implicitly compare to previous machine translation approaches like (Espana-Bonet et al., 2017; Schwenk & Douze, 2017; Artetxe & Schwenk, 2018) in Appendix A where we explore different variations of training LSTM sequence-to-sequence models. We find that our translation baselines reported in the tables below (both LSTM and Transformer) outperform the architectures from these works due to using the Attention and Logit methods mentioned in Section 2 , demonstrating that our baselines represent, or even over-represent, the state-of-the-art for machine translation approaches.

**BGT Ablations**    Lastly, we compare to ablations of our model to better understand the benefits of parallel data, language-specific variables, the KL loss term, and how much we gain from the more conventional translation baselines.

- ENGLISHAE: English autoencoder on the English side of our `en-fr` data.
- ENGLISHVAE: English variational autoencoder on the English side of our `en-fr` data.
- ENGLISHTRANS: Translation from `en` to `fr`.
- BILINGUALTRANS: Translation from both `en` to `fr` and `fr` to `en` where the encoding parameters are shared but each language has its own decoder.
- BGT W/O LANGVARS: A model similar to BILINGUALTRANS, but it includes a prior over the embedding space and therefore a KL loss term. This model differs from BGT since it does not have any language-specific variables.
- BGT W/O PRIOR: Follows the same architecture as BGT, but without the priors and KL loss term.

---

[2]Note that in all experiments using BERT, including Sentence-BERT, the large, uncased version is used.

[3]Most work evaluating accuracy on STS tasks has averaged the Pearson's $r$ over each individual dataset for each year of the STS competition. However, Reimers & Gurevych (2019) computed Spearman's $\rho$ over concatenated datasets for each year of the STS competition. To be consistent with previous work, we re-ran their model and calculated results using the standard method, and thus our results are not the same as those reported Reimers & Gurevych (2019).

## 4.2 EXPERIMENTAL SETTINGS

The training data for our models is a mixture of OpenSubtitles 2018[4] `en-fr` data and `en-fr` Gigaword[5] data. To create our dataset, we combined the complete corpora of each dataset and then randomly selected 1,000,000 sentence pairs to be used for training with 10,000 used for validation. We use `sentencepiece` (Kudo & Richardson, 2018) with a vocabulary size of 20,000 to segment the sentences, and we chose sentence pairs whose sentences are between 5 and 100 tokens each.

In designing the model architectures for the encoders and decoders, we experimented with Transformers and LSTMs. Due to better performance, we use a 5 layer Transformer for each of the encoders and a single layer decoder for each of the decoders. This design decision was empirically motivated as we found using a larger decoder was slower and worsened performance, but conversely, adding more encoder layers improved performance. More discussion of these trade-offs along with ablations and comparisons to LSTMs are included in Appendix C.

For all of our models, we set the dimension of the embeddings and hidden states for the encoders and decoders to 1024. Since we experiment with two different architectures,[6] we follow two different optimization strategies. For training models with Transformers, we use Adam (Kingma & Ba, 2014) with $\beta_1 = 0.9$, $\beta_2 = 0.98$, and $\epsilon = 10^{-8}$. We use the same learning rate schedule as (Vaswani et al., 2017), i.e., the learning rate increases linearly for 4,000 steps to $5 \times 10^{-4}$, after which it is decayed proportionally to the inverse square root of the number of steps. For training the LSTM models, we use Adam with a fixed learning rate of 0.001. We train our models for 20 epochs.

For models incorporating a translation loss, we used label smoothed cross entropy (Szegedy et al., 2016; Pereyra et al., 2017) with $\epsilon = 0.1$. For ENGLISHVAE, BGT and BILINGUALTRANS, we anneal the KL term so that it increased linearly for $2^{16}$ updates, which robustly gave good results in preliminary experiments. We also found that in training BGT, combining its loss with the BILINGUALTRANS objective during training of both models increased performance, and so this loss was summed with the BGT loss in all of our experiments. We note that this doesn't affect our claim of BGT being a generative model, as this loss is only used in a multi-task objective at training time, and we calculate the generation probabilities according to standard BGT at test time.

Lastly, in Appendix B, we illustrate that it is crucial to train the Transformers with large batch sizes. Without this, the model can learn the goal task (such as translation) with reasonable accuracy, but the learned semantic embeddings are of poor quality until batch sizes approximately reach 25,000 tokens. Therefore, we use a maximum batch size of 50,000 tokens in our ENGLISHTRANS, BILINGUALTRANS, and BGT W/O PRIOR, experiments and 25,000 tokens in our BGT W/O LANGVARS and BGT experiments.

## 4.3 EVALUATION

Our primary evaluation are the 2012-2016 SemEval Semantic Textual Similarity (STS) shared tasks (Agirre et al., 2012; 2013; 2014; 2015; 2016), where the goal is to accurately predict the degree to which two sentences have the same meaning as measured by human judges. The evaluation metric is Pearson's $r$ with the gold labels.

Secondly, we evaluate on *Hard STS*, where we combine and filter the STS datasets in order to make a more difficult evaluation. We hypothesize that these datasets contain many examples where their gold scores are easy to predict by either having similar structure and word choice and a high score or dissimilar structure and word choice and a low score. Therefore, we split the data using symmetric word error rate (SWER),[7] finding sentence pairs with low SWER and low gold scores as well as sentence pairs with high SWER and high gold scores. This results in two datasets, Hard+ which have SWERs in the bottom 20% of all STS pairs and whose gold label is between 0 and 1,[8] and

---

[4]`http://opus.nlpl.eu/OpenSubtitles.php`

[5]`https://www.statmt.org/wmt10/training-giga-fren.tar`

[6]We use LSTMs in our ablations.

[7]We define symmetric word error rate for sentences $s_1$ and $s_2$ as $\frac{1}{2} WER(s_1, s_2) + \frac{1}{2} WER(s_2, s_2)$, since word error rate (WER) is an asymmetric measure.

[8]STS scores are between 0 and 5.

| Data | Sentence 1 | Sentence 2 | Gold Score |
|---|---|---|---|
| Hard+ | Other ways are needed. | It is necessary to find other means. | 4.5 |
| Hard- | How long can you keep chocolate in the freezer? | How long can I keep bread dough in the refrigerator? | 1.0 |
| Negation | It's not a good idea. | It's a good idea to do both. | 1.0 |

Table 1: Examples from our *Hard STS* dataset and our negation split. The sentence pair in the first row has dissimilar structure and vocabulary yet a high gold score. The second sentence pair has similar structure and vocabulary and a low gold score. The last sentence pair contains negation, where there is a *not* in Sentence 1 that causes otherwise similar sentences to have low semantic similarity.

| Model | Semantic Textual Similarity (STS) | | | | | | | | |
|---|---|---|---|---|---|---|---|---|---|
| | 2012 | 2013 | 2014 | 2015 | 2016 | **Avg.** | Hard+ | Hard- | **Avg.** |
| BERT (CLS) | 33.2 | 29.6 | 34.3 | 45.1 | 48.4 | 38.1 | 7.8 | 12.5 | 10.2 |
| BERT (Mean) | 48.8 | 46.5 | 54.0 | 59.2 | 63.4 | 54.4 | 3.1 | 24.1 | 13.6 |
| Infersent | 61.1 | 51.4 | 68.1 | 70.9 | 70.7 | 64.4 | 4.2 | 29.6 | 16.9 |
| GenSen | 60.7 | 50.8 | 64.1 | 73.3 | 66.0 | 63.0 | **24.2** | 6.3 | 15.3 |
| USE | 61.4 | 59.0 | 70.6 | 74.3 | 73.9 | 67.8 | 16.4 | 28.1 | 22.3 |
| Sentence-BERT | 66.9 | **63.2** | 74.2 | 77.3 | 72.8 | 70.9 | 23.9 | 3.6 | 13.8 |
| SP | 68.4 | 60.3 | 75.1 | 78.7 | 76.8 | 71.9 | 19.1 | 29.8 | 24.5 |
| BiLSTM | 67.9 | 56.4 | 74.5 | 78.2 | 75.9 | 70.6 | 18.5 | 23.2 | 20.9 |
| EnglishAE | 60.2 | 52.7 | 68.6 | 74.0 | 73.2 | 65.7 | 15.7 | 36.0 | 25.9 |
| EnglishVAE | 59.5 | 54.0 | 67.3 | 74.6 | 74.1 | 65.9 | 16.8 | 42.7 | 29.8 |
| EnglishTrans | 66.5 | 60.7 | 72.9 | 78.1 | 78.3 | 71.3 | 18.0 | 47.2 | 32.6 |
| BilingualTrans | 67.1 | 61.0 | 73.3 | 78.0 | 77.8 | 71.4 | 20.0 | **48.2** | 34.1 |
| BGT w/o LangVars | 68.3 | 61.3 | 74.5 | 79.0 | 78.5 | 72.3 | 24.1 | 46.8 | **35.5** |
| BGT w/o Prior | 67.6 | 59.8 | 74.1 | 78.4 | 77.9 | 71.6 | 17.9 | 45.5 | 31.7 |
| BGT | **68.9** | 62.2 | **75.9** | **79.4** | **79.3** | **73.1** | 22.5 | 46.6 | 34.6 |

Table 2: Results of our models and models from prior work. The first six rows are pretrained models from the literature, the next two rows are strong baselines trained on the same data as our models, and the last seven rows include model ablations and BGT, our final model. We show results, measured in Pearson's $r \times 100$, for each year of the STS tasks 2012-2016 and our two *Hard STS* datasets.

Hard- where the SWERs are in the top 20% of the gold scores are between 4 and 5. We also evaluate on a split where negation was likely present in the example.[9] Examples are shown in Table 1.

Lastly, we evaluate on STS in `es` and `ar` as well as cross-lingual evaluations for `en-es`, `en-ar`, and `en-tr`. We use the datasets from SemEval 2017 (Cer et al., 2017). For this setting, we train BilingualTrans and BGT on 1 million examples from `en-es`, `en-ar`, and `en-tr` OpenSubtitles 2018 data.

## 4.4 Results

The results on the STS and *Hard STS* are shown in Table 2.[10] From the results, we see that BGT has the highest overall performance. It does especially well compared to prior work on the two *Hard STS* datasets.

We show further difficult splits in Table 3, including a negation split, beyond those used in *Hard STS* and compare the top two performing models in the STS task from Table 2. We also show easier splits in the bottom of the table.

From these results, we see that both positive examples that have little shared vocabulary and structure and negative examples with significant shared vocabulary and structure benefit significantly from using a deeper architecture. Similarly, examples where negation occurs also benefit from our deeper model. These examples are difficult because more than just the identity of the words is needed to

---

[9]We selected examples for the negation split where one sentence contained *not* or *'t* and the other did not.

[10]We obtained values for STS 2012-2016 from prior works using SentEval (Conneau & Kiela, 2018). Note that we include all datasets for the 2013 competition, including SMT, which is not included in SentEval.

| Data Split | $n$ | BGT | SP |
|---|---|---|---|
| All | 13,023 | **75.3** | 74.1 |
| Negation | 705 | **73.1** | 68.7 |
| Bottom 20% SWER, label $\in [0, 2]$ | 404 | **63.6** | 54.9 |
| Bottom 10% SWER, label $\in [0, 1]$ | 72 | **47.1** | 22.5 |
| Top 20% SWER, label $\in [3, 5]$ | 937 | **20.0** | 14.4 |
| Top 10% SWER, label $\in [4, 5]$ | 159 | **18.1** | 10.8 |
| Top 20% WER, label $\in [0, 2]$ | 1380 | **51.5** | 49.9 |
| Bottom 20% WER, label $\in [3, 5]$ | 2079 | **43.0** | 42.2 |

Table 3: Performance, measured in Pearson's $r \times 100$, for different data splits of the STS data. The first row shows performance across all unique examples, the next row shows the negation split, and the last four rows show difficult examples filtered symmetric word error rate (SWER). The last two rows show relatively easy examples according to SWER.

| Model | es-es | ar-ar | en-es | en-ar | en-tr |
|---|---|---|---|---|---|
| BILINGUALTRANS | 83.4 | 72.6 | 64.1 | 37.6 | 59.1 |
| BGT W/O LANGVARS | 81.7 | 72.8 | 72.6 | 73.4 | 74.8 |
| BGT W/O PRIOR | 84.5 | 73.2 | 68.0 | 66.5 | 70.9 |
| BGT | **85.7** | **74.9** | **75.6** | **73.5** | **74.9** |

Table 4: Performance measured in Pearson's $r \times 100$, on the SemEval 2017 STS task on the es-es, ar-ar, en-es, en-ar, and en-tr datasets.

determine the relationship of the two sentences, and this is something that SP is not equipped for since it is unable to model word order. The bottom two rows show *easier* examples where positive examples have high overlap and low SWER and vice versa for negative examples. Both models perform similarly on this data, with the BGT model having a small edge consistent with the overall gap between these two models.

Lastly, in Table 4, we show the results of STS evaluations in es and ar and cross-lingual evaluations for en-es, en-ar, and en-tr. From these results, we see that BGT has the best performance across all datasets, however the performance is significantly stronger than the BILINGUALTRANS and BGT W/O PRIOR baselines in the cross-lingual setting. Since BGT W/O LANGVARS also has significantly better performance on these tasks, most of this gain seems to be due to the prior have a regularizing effect. However, BGT outperforms BGT W/O LANGVARS overall, and we hypothesize that the gap in performance between these two models is due to BGT being able to strip away the language-specific information in the representations with its language-specific variables, allowing for the semantics of the sentences to be more directly compared.

## 5 ANALYSIS

We next analyze our BGT model by examining what elements of syntax and semantics the language and semantic variables capture relative both to each-other and to the sentence embeddings from the BILINGUALTRANS models. We also analyze how the choice of language and its lexical and syntactic distance from English affects the semantic and syntactic information captured by the semantic and language-specific encoders. Finally, we also show that our model is capable of sentence generation in a type of *style transfer*, demonstrating its capabilities as a generative model.

### 5.1 STS

We first show that the language variables are capturing little semantic information by evaluating the learned English language-specific variable from our BGT model on our suite of semantic tasks. The results in Table 5 show that these encoders perform closer to a random encoder than the semantic encoder from BGT. This is consistent with what we would expect to see if they are capturing extraneous language-specific information.

### 5.2 PROBING

We probe our BGT semantic and language-specific encoders, along with our BILINGUALTRANS encoders as a baseline, to compare and contrast what aspects of syntax and semantics they are

| Model | Semantic Textual Similarity (STS) | | | | | | |
|---|---|---|---|---|---|---|---|
| | 2012 | 2013 | 2014 | 2015 | 2016 | Hard+ | Hard- |
| Random Encoder | 51.4 | 34.6 | 52.7 | 52.3 | 49.7 | 4.8 | 17.9 |
| English Language Encoder | 44.4 | 41.7 | 53.8 | 62.4 | 61.7 | 15.3 | 26.5 |
| Semantic Encoder | **68.9** | **62.2** | **75.9** | **79.4** | **79.3** | **22.5** | **46.6** |

Table 5: STS performance on the 2012-2016 datasets and our *STS Hard* datasets for a randomly initialized Transformer, the trained English language-specific encoder from BGT, and the trained semantic encoder from BGT. Performance is measured in Pearson's $r \times 100$.

| Lang. | Model | STS | S. Num. | O. Num. | Depth | Top Con. | Word | Len. | P. Num. | P. First | Gend. |
|---|---|---|---|---|---|---|---|---|---|---|---|
| fr | BILINGUALTRANS | 71.2 | 78.0 | 76.5 | 28.2 | 65.9 | **80.2** | 74.0 | 56.9 | 88.3 | 53.0 |
| | Semantic Encoder | **72.4** | **84.6** | **80.9** | **29.7** | **70.5** | 77.4 | 73.0 | 60.7 | 87.9 | 52.6 |
| | en Language Encoder | 56.8 | 75.2 | 72.0 | 28.0 | 63.6 | 65.4 | **80.2** | 65.3 | 92.2 | - |
| | fr Language Encoder | - | - | - | - | - | - | - | - | - | **56.5** |
| es | BILINGUALTRANS | 70.5 | 84.5 | 82.1 | 29.7 | 68.5 | **79.2** | 77.7 | 63.4 | 90.1 | 54.3 |
| | Semantic Encoder | **72.1** | **85.7** | **83.6** | **32.5** | **71.0** | 77.3 | 76.7 | 63.1 | 89.9 | 52.6 |
| | en Language Encoder | 55.8 | 75.7 | 73.7 | 29.1 | 63.9 | 63.3 | **80.2** | 64.2 | 92.7 | - |
| | es Language Encoder | - | - | - | - | - | - | - | - | - | **54.7** |
| ar | BILINGUALTRANS | 70.2 | 77.6 | 74.5 | 28.1 | 67.0 | **77.5** | 72.3 | 57.5 | 89.0 | - |
| | Semantic Encoder | **70.8** | **81.9** | **80.8** | **32.1** | **71.7** | 71.9 | 73.3 | 61.8 | 88.5 | - |
| | en Language Encoder | 58.9 | 76.2 | 73.1 | 28.4 | 60.7 | 71.2 | **79.8** | 63.4 | 92.4 | - |
| tr | BILINGUALTRANS | 70.7 | 78.5 | 74.9 | 28.1 | 60.2 | **78.4** | 72.1 | 54.8 | 87.3 | - |
| | Semantic Encoder | **72.3** | **81.7** | **80.2** | **30.6** | **66.0** | 75.2 | 72.4 | 59.3 | 86.7 | - |
| | en Language Encoder | 57.8 | 77.3 | 74.4 | 28.3 | 63.1 | 67.1 | **79.7** | 67.0 | 92.5 | - |
| ja | BILINGUALTRANS | 71.0 | 66.4 | 64.6 | 25.4 | 54.1 | **76.0** | 67.6 | 53.8 | 87.8 | - |
| | Semantic Encoder | **71.9** | 68.0 | 66.8 | 27.5 | 58.9 | 70.1 | 68.7 | 52.9 | 86.6 | - |
| | en Language Encoder | 60.6 | **77.6** | **76.4** | **28.0** | **64.6** | 70.0 | **80.4** | 62.8 | 92.0 | - |

Table 6: Average STS performance for the 2012-2016 datasets, measured in Pearson's $r \times 100$, followed by probing results on predicting number of subjects, number of objects, constituent tree depth, top constituent, word content, length, number of punctuation marks, the first punctuation mark, and whether the articles in the sentence are the correct gender. All probing results are measured in accuracy $\times 100$.

learning relative to each other across five languages with various degrees of similarity with English. All models are trained on the OpenSubtitles 2018 corpus. We use the datasets from (Conneau et al., 2018) for semantic tasks like number of subjects and number of objects, and syntactic tasks like tree depth, and top constituent. Additionally, we include predicting the word content and sentence length. We also add our own tasks to validate our intuitions about punctuation and language-specific information. In the first of these, *punctuation number*, we train a classifier to predict the number of punctuation marks[11] in a sentence. To make the task more challenging, we limit each label to have at most 20,000 examples split among training, validation, and testing data.[12] In the second task, *punctuation first*, we train a classifier to predict the identity of the first punctuation mark in the sentence. In our last task, *gender*, we detect examples where the gender of the articles in the sentence is incorrect in French of Spanish. To create an incorrect example, we switch articles from {le, la, un, une} for French and {el, la, los, las} for Spanish, with their (indefinite or definite for French and singular or plural for Spanish) counterpart with the opposite gender. This dataset was balanced so random chances gives 50% on the testing data. All tasks use 100,000 examples for training and 10,000 examples for validation and testing. The results of these experiments are shown in Table 6.

These results show that the source separation is effective - stylistic and language-specific information like length, punctuation and language-specific gender information are more concentrated in the language variables, while word content, semantic and syntactic information are more concentrated in the semantic encoder. The choice of language is also seen to be influential on what these encoders are capturing. When the languages are closely related to English, like in French and Spanish, the performance difference between the semantic and English language encoder is larger for word content, subject number, object number than for more distantly related languages like Arabic and

---

[11]Punctuation were taken from the set { ' ! " # $ % & \' ( ) * + , − . / : ; < = > ? @ [ ] ^ _ ` { — }⁵ . }.

[12]The labels are from 1 punctuation mark up to 10 marks with an additional label consolidating 11 or more marks.

Turkish. In fact, word content performance is directly tied to how well the alphabets of the two languages overlap. This relationship matches our intuition, because lexical information will be cheaper to encode in the semantic variable when it is shared between the languages. Similarly for the tasks of length, punctuation first, and punctuation number, the gap in performance between the two encoders also grows as the languages become more distant from English. Lastly, the gap on STS performance between the two encoders shrinks as the languages become more distant, which again is what we would expect, as the language-specific encoders are forced to capture more information.

Japanese is an interesting case in these experiments, where the English language-specific encoder outperforms the semantic encoder on the semantic and syntactic probing tasks. Japanese is a very distant language to English both in its writing system and in its sentence structure (it is an SOV language, where English is an SVO language). However, despite these difference, the semantic encoder strongly outperforms the English language-specific encoder, suggesting that the underlying meaning of the sentence is much better captured by the semantic encoder.

### 5.3 GENERATION AND STYLE TRANSFER

| | |
|---|---|
| Source | you know what i've seen? |
| Style | he said, "since when is going fishing" had anything to do with fish?" |
| Output | he said, "what is going to do with me since i saw you?" |
| Source | guys, that was the tech unit. |
| Style | is well, "capicci" ... |
| Output | is that what, "technician"? |
| Source | the pay is no good, but it's money. |
| Style | do we know cause of death? |
| Output | do we have any money? |
| Source | we're always doing stupid things. |
| Style | all right listen, i like being exactly where i am, |
| Output | all right, i like being stupid, but i am always here. |

Table 7: *Style transfer* generations from our learned BGT model. *Source* refers to the sentence fed into the semantic encoder, *Style* refers to the sentence fed into the English language-specific encoder, and *Output* refers to the text generated by our model.

In this section, we qualitatively demonstrate the ability of our model to generate sentences. We focus on a *style-transfer* task where we have original seed sentences from which we calculate our semantic vector $z_{sem}$ and language specific vector $z_{en}$. Specifically, we feed in a *Source* sentence into the semantic encoder to obtain $z_{sem}$, and another *Style* sentence into the English language-specific encoder to obtain $z_{en}$. We then generate a new sentence using these two latent variables. This can be seen as a type of style transfer where we expect the model to generate a sentence that has the semantics of the *Source* sentence and the style of the *Style* sentence. We use our `en-fr` BGT model from Table 6 and show some examples in Table 7. All input sentences are from held-out `en-fr` OpenSubtitles data. From these examples, we see further evidence of the role of the semantic and language-specific encoders, where most of the semantics (e.g. topical word such as *seen* and *tech* in the *Source* sentence) are reflected in the output, but length and structure are more strongly influenced by the language-specific encoder.

## 6 CONCLUSION

We propose Bilingual Generative Transformers, a model that uses parallel data to learn to perform *source separation* of common semantic information between two languages from language-specific information. We show that the model is able to accomplish this source separation through probing tasks and text generation in a style-transfer setting. We find that our model bests all baselines on semantic similarity tasks, with the largest gains coming from a new challenge we propose as *Hard STS*, designed to foil methods approximating semantic similarity as word overlap. We also find our model to be especially effective on cross-lingual semantic similarity, due to its stripping away of language-specific information allowing for the underlying semantics to be more directly compared. In future work, we will explore generalizing this approach to the multilingual setting.

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

## A    Location of Sentence Embedding in Decoder for Learning Representations

As mentioned in Section 2, we experimented with 4 ways to incorporate the sentence embedding into the decoder: Word, Hidden, Attention, and Logit. We also experimented with combinations of these 4 approaches. We evaluate these embeddings on the STS tasks and show the results, along with the time to train the models 1 epoch in Table 8.

For these experiments, we train a single layer bidirectional LSTM (BiLSTM) ENGLISHTRANS model with embedding size set to 1024 and hidden states set to 512 dimensions (in order to be roughly equivalent to our Transformer models). To form the sentence embedding in this variant, we mean pool the hidden states for each time step. The cell states of the decoder are initialized to the zero vector.

| Architecture | STS | Time (s) |
|---|---|---|
| BiLSTM (Hidden) | 54.3 | 1226 |
| BiLSTM (Word) | 67.2 | 1341 |
| BiLSTM (Attention) | 68.8 | 1481 |
| BiLSTM (Logit) | 69.4 | 1603 |
| BiLSTM (Word + Hidden) | 67.3 | 1377 |
| BiLSTM (Word + Hidden + Attention) | 68.3 | 1669 |
| BiLSTM (Word + Hidden + Logit) | 69.1 | 1655 |
| BiLSTM (Word + Hidden + Attention + Logit) | 68.9 | 1856 |

Table 8:   Results for different ways of incorporating the sentence embedding in the decoder for a BiLSTM on the Semantic Textual Similarity (STS) datasets, along with the time taken to train the model for 1 epoch. Performance is measured in Pearson's $r \times 100$.

From this analysis, we see that the best performance is achieved with Logit, when the sentence embedding is place just prior to the softmax. The performance is much better than Hidden or Hidden+Word used in prior work. For instance, recently (Artetxe & Schwenk, 2018) used the Hidden+Word strategy in learning multilingual sentence embeddings.

### A.1    VAE Training

We also found that incorporating the latent code of a VAE into the decoder using the Logit strategy increases the mutual information while having little effect on the log likelihood. We trained two LSTM VAE models following the settings and aggressive training strategy in (He et al., 2019), where one LSTM model used the Hidden strategy and the other used the Hidden + Logit strategy. We trained the models on the `en` side of our `en-fr` data. We found that the mutual information increased form 0.89 to 2.46, while the approximate negative log likelihood, estimated by importance weighting, increased slightly from 53.3 to 54.0 when using Logit.

## B    Relationship Between Batch Size and Performance for Transformer and LSTM

It has been observed previously that the performance of Transformer models is sensitive to batch size Popel & Bojar (2018) . We found this to be especially true when training sequence-to-sequence models to learn sentence embeddings. Figure 3 shows plots of the average 2012-2016 STS performance of the learned sentence embedding as batch size increases for both the BiLSTM and Transformer. Initially, at a batch size of 2500 tokens, sentence embeddings learned are worse than random, even though validation perplexity does decrease during this time. Performance rises as batch size increases up to around 100,000 tokens. In contrast, the BiLSTM is more robust to batch size, peaking much earlier around 25,000 tokens, and even degrading at higher batch sizes.

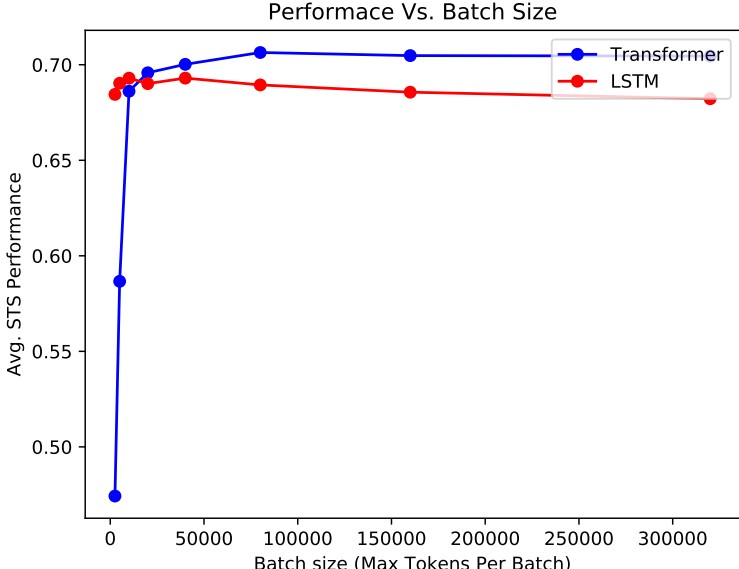

Figure 3: The relationship between average performance for each year of the STS tasks 2012-2016 (Pearson's $r \times 100$) and batch size (maximum number of words per batch).

| Architecture | STS | Time (s) |
|---|---|---|
| Transformer (5L/1L) | 70.3 | 1767 |
| Transformer (3L/1L) | 70.1 | 1548 |
| Transformer (1L/1L) | 70.0 | 1244 |
| Transformer (5L/5L) | 69.8 | 2799 |

Table 9: Results on the Semantic Textual Similarity (STS) datasets for different configurations of ENGLISHTRANS, along with the time taken to train the model for 1 epoch. (XL/YL) means X layers were used in the encoder and Y layers in the decoder. Performance is measured in Pearson's $r \times 100$.

## C    MODEL ABLATIONS

In this section, we vary the number of layers in the encoder and decoder in BGT W/O PRIOR. We see that performance increases as the number of encoder layers increases, and also that a large decoder hurts performance, allowing us to save training time by using a single layer. These results can be compared to those in Table 9 showing that Transformers outperform BiLSTMS in these experiments.

## D    CLASSIFICATION EXPERIMENTS

To explore our embeddings in more detail, we evaluated them on the Quora Question Pairs dataset[13] (QQP). This is a paraphrase classification task, which is also part of GLUE (Wang et al., 2018). Since the test set is private, we deviated slightly from the standard evaluation protocol and split the development set into two halves of 20,215 examples each – one half for model selection and the other for evaluation. We evaluated in two ways, cosine, where we score all pairs with cosine similarity and then find the threshold that gives the best accuracy, and logistic regression where we use logistic regression. Its worth noting that the pretrained baseline models on this task were directly trained to produce the feature set used by the downstream classifier, while our embeddings are trained without this supervision. They also tend to have larger dimensions which also gives them an advantage which is discussed in more detail in (Wieting & Kiela, 2019). The results are shown in Table 10 and show that our BGT model outperforms the baseline models, SP, ENGLISHTRANS,

---

[13]data.quora.com/First-Quora-Dataset-Release-Question-Pairs

| Model | Dim | QQP (cosine) | QQP (logistic regression) |
|---|---|---|---|
| BERT (CLS) | 4096 | 65.7 | 77.2 |
| BERT (Mean) | 4096 | 68.9 | 79.3 |
| Infersent | 4096 | 69.3 | 79.9 |
| GenSen | 4096 | 68.1 | 80.9 |
| USE | 512 | **75.8** | 78.9 |
| Sentence-BERT | 1024 | 74.5 | **81.0** |
| SP | 1024 | 69.6 | 76.0 |
| ENGLISHTRANS | 1024 | 69.5 | 77.3 |
| BGT W/O PRIOR | 1024 | 69.4 | 77.1 |
| BGT | 1024 | 69.9 | 77.5 |

Table 10: Results on the Quora Question Pairs (QQP) datasets for prior work, our baselines and our BGT model using two classification strategies. Performance is measured in accuracy $\times 100$.

and BILINGUALTRANS for both evaluations, and compares favorably to the pretrained models when evaluated using cosine similarity scores. The only models which perform better are USE which was trained on Quora data in an unsupervised way and Sentence-BERT which uses BERT. Our models are not as strong when using classification for final predictions. This indicates that the embeddings learned by our approach may be most useful when no downstream training is possible – though semi-supervised objectives that consider the downstream task might aid our approach, like the baselines, if downstream training is the goal.

