# OpenReview forum: "A Bilingual Generative Transformer for Semantic Sentence Embedding"
_ICLR.cc/2020/Conference — Reject_

### Official Review · AnonReviewer2 · 2019-10-23
**Official Blind Review #2**

**Rating:** 3

**Review:**

This paper presents a bilingual generative model for sentence embedding based variational probabilistic framework. By separating a common latent variable from language-specific latent variables, the model is able to capture what's in common between parallel bilingual sentences and language-specific semantics. Experimental results show that the proposed model is able to produce sentence embeddings that reach higher correlation scores with human judgments on Semantic Textual Similarity tasks than previous models such as BERT.

Strength: 1) the idea of separating common semantics and language-specific semantics in the latent space is pretty neat; 2) the writing is very clear and easy to follow; 3) the authors explore four approaches to use the latent vectors and four approaches to merge semantic vectors, makes the final choices reasonable.

Weakness:

1) Experiments:

My major concern is the fairness of the experiments.  The authors compare their model with many state-of-the-art models that could produce sentence embeddings. However, how they produce the sentence embeddings with existing models is not convincing. For example, why using the hidden states of the last four layers of BERT? Moreover, the proposed model is trained with parallel bilingual data, while the BERT model in comparison is monolingual.  Also, the proposed deep variational model is close to an auto-encoder framework. You can also train a bilingual encoder-decoder transformer model (perhaps with pre-trained BERT parameters) with auto-encoder objective using the same parallel data set.  It seems to be a more comparable model to me.

Although the proposed model is based on variational framework, there's no comparison with previous neural variational models that learn encodings of texts as well such as https://arxiv.org/abs/1511.06038.

2) Ablation study and analysis

I really like the idea of separating common semantic latent variables with language-specific latent variables.  However, I expected to see more analysis or experimental results to show why it is better than a monolingual variational sentence embedding framework.

**Experience Assessment:**

I have read many papers in this area.

**Review Assessment: Checking Correctness Of Derivations And Theory:**

I carefully checked the derivations and theory.

**Review Assessment: Checking Correctness Of Experiments:**

I carefully checked the experiments.

**Review Assessment: Thoroughness In Paper Reading:**

I read the paper thoroughly.

---

> ### Author Response · Authors · 2019-11-15
> **Response to Reviewer #2**
>
> Thanks you for the review and comments! We have addressed them below:
>
> "compare their model with many state-of-the-art models that could produce sentence embeddings. However, how they produce the sentence embeddings with existing models is not convincing. For example, why using the hidden states of the last four layers of BERT?"
>
> Thanks for drawing our attention to this point of confusion! For all the approaches we compare with, apart from BERT, we produce the sentence embeddings using the exact implementations described in the corresponding paper using the released code. For BERT, as far as we know, there is no set way of creating sentence embeddings for directly computing sentence similarity. But, to help address your question, we have added comparisons to Sentence-BERT [1] (a model that fine-tunes BERT on SNLI+MNLI data) to give a stronger baseline for a BERT-based approach. In results, we find that our proposed models substantially outperform this fine-tuned BERT-based approach. While this is not the main evaluation in our paper, we believe it indicates that BERT is not appropriate for this type of unsupervised sentence similarity task without substantial modification / further innovation.  Finally, we want to briefly clarify: our reasoning behind concatenating the last four layers of BERT is based on the positive results in [1] and that this approach was recommended in the original BERT paper [2].
>
> "However, I expected to see more analysis or experimental results to show why it is better than a monolingual variational sentence embedding framework."
>
> We completely agree that further comparisons of this type are of general interest. Thanks for the comment! We have since added more analysis to the paper in the form of probing experiments and ablations showing the benefits of each component of our model. We also added a monolingual autoencoder and VAE to better understand  the impact the parallel data has on the performance of our models (in Table 2). We hope that you find the analysis and new experiments helpful.
>
> [1] Nils Reimers and Iryna Gurevych. Sentence-bert: Sentence embeddings using siamese bertnetworks. EMNLP, 2019.
> [2] Jacob Devlin, Ming-Wei Chang, Kenton Lee, and Kristina Toutanova. Bert: Pre-training of deep bidirectional transformers for language understanding. NAACL, 2019.

---

### Official Review · AnonReviewer1 · 2019-10-23
**Official Blind Review #1**

**Rating:** 6

**Review:**

This paper addresses the problem of constructing a sentence embedding using a generative transformer model which encodes semantic aspects and language-specific aspect separately. They use transformers to encode and decode sentence embedding, and the objective reconstructs input with a latent variables (language variables for each language and semantic language).  These latent variables are sampled from multivariate Gaussian prior, and the learning uses evidence lower bound (ELBO) for variational approximation of the joint distribution of latent variables and input.

The method is evaluated on two tasks: sentence similarity task and machine translation evaluation metric tasks. Both tasks evaluates how similar are two sequences, and the metric is correlation score with score’s from human judge. The model shows promising results on the first task, but weaker results on the second task, especially when compared against pretty naively built sentence embedding from BERT model. I’m not expert in sentence embedding literature, so a bit tricky to evaluate, but baselines seem strong and experimental results on semantic textual similarity task.

In terms of evaluation, I appreciated how they defined harder subset of the evaluation dataset and showed a larger improvements on those portions of the dataset. The The paper also includes analysis on what is captured by their language-specific latent vector and semantic latent vector. While I’m not totally convinced this distinction between language-specific characteristics and semantics of the sentence, it makes it easier to understand what’s going on in the model.

One of my question is, why not test this method in more popular benchmark such as MNLI or other classification tasks? MNLI evaluates how each sentence pair relates to one another, thus would be a good benchmark for sentence embeddings as well. Having to encode all the information about a sentence into a single vector will make these sentence embedding model weaker than other models which can do cross sentence attentions and etc, but I think that’s the genuine limitation of sentence embedding research and has to be clarified as such. I recommend discussing and clarifying these points.

I’m a bit unclear how these sentence embeddings are translated into a score that decides the degree to which sentences have the same meaning. Is it just cosine similarity of two sentence embedding vectors?

While the purpose of these references is to generate sentences instead of building a sentence embedding, the method is related and comparison and discussion would be worthwhile.

Generating Sentences from a Continuous Space
Samuel R. Bowman, Luke Vilnis, Oriol Vinyals, Andrew M. Dai, Rafal Jozefowicz, Samy Bengio
https://arxiv.org/abs/1511.06349
Toward Controlled Generation of Text
Zhiting Hu, Zichao Yang, Xiaodan Liang, Ruslan Salakhutdinov, Eric P. Xing
https://arxiv.org/abs/1703.00955


Comments & Questions:
- Methods using a large amount of unsupervised monolingual data shows very strong performance in a panoply of NLP tasks these days. If I understand correctly, this model is constrained by the amount of bitext — some analysis on this would be interesting.
- Figure 1 mentions about “Section 3, 4” but I don’t think they are correct references?
- BERT baseline seemed not to allow fine-tuning of the LM parameters. I think this makes the baseline significantly weaker?
- It seems odd that only English semantic encoder is used to downstream application.
- Does table 3 covers all the data? What proportion of the data is covered by each row?
- Given the similarity of English and French, I’m not sure how “language-specific” such latent vectors are. It would be much more interesting analysis if it studies distant language pairs.

**Experience Assessment:**

I have read many papers in this area.

**Review Assessment: Checking Correctness Of Derivations And Theory:**

I assessed the sensibility of the derivations and theory.

**Review Assessment: Checking Correctness Of Experiments:**

I carefully checked the experiments.

**Review Assessment: Thoroughness In Paper Reading:**

I read the paper at least twice and used my best judgement in assessing the paper.

---

> ### Author Response · Authors · 2019-11-15
> **Response to Reviewer #1 (Part 1)**
>
> Thank you for the feedback! We address your comments below.
>
> "While I’m not totally convinced this distinction between language-specific characteristics and semantics of the sentence, it makes it easier to understand what’s going on in the model."
>
> We hope that the experiments with the model ablations and the newly added probing experiments make this distinction more convincing, clearer and better motivated.
>
> "One of my question is, why not test this method in more popular benchmark such as MNLI or other classification tasks?"
>
> Thank you for this question. There are a few reasons for this (and we have added an experiment in Appendix D). The main reason for not comparing on MNLI specifically is that all of the pretrained baseline methods are created by training on SNLI + MNLI (either in full as in Infersent and Sentence-BERT, or are one of the objectives as in USE or GenSen), so they have a big advantage since they use direct supervision.
>
> However, we agree that evaluation on additional datasets and tasks is of general interest -- and further, since different tasks use sentence embeddings in different ways, how our embeddings are most useful in practice is worth further investigation. Throughout STS, cosine distance between embeddings is used to define similarity. But in some related tasks, regression models are fit on top of embeddings, or classifiers are trained on top of similarity scores, etc.., in order to make task predictions. Are our embeddings specially suited for cosine distance evaluations (which requires no further training)? Do they also work when supervised models are trained to predict similarity scores based on the embeddings?
>
> To explore these questions, we have added the Quora Question Pairs dataset (QQP) to our evaluations. QQP is a paraphrase classification task that is also part of GLUE [1]. Since the test set of QQP is private, we deviated slightly from the standard evaluation protocol and split the development set into 2 halves of 20,215 examples each -- one half for model selection and the other for evaluation. We evaluated in two ways: cosine similarity (score all pairs with cosine similarity and then find the threshold that gives the best accuracy) and classification by training a logistic regression model on top of embeddings. It’s worth noting that the pretrained baseline models on this task were directly trained to produce the feature set used by the downstream classifier, while our embeddings are trained without this supervision. We describe the results in Appendix D and show that our model outperforms our baseline models, SP, EnglishTranslation, and BilingualTranslation for both evaluations, and compares favorably to the pretrained models when using cosine (only behind USE which was trained on Quora data in an unsupervised way and Sentence-BERT which uses BERT), but it isn’t as strong when using classification for final predictions. This indicates that the embeddings learned by our approach may be most useful when no downstream training is possible -- though semi-supervised objectives that consider the downstream task might aid our approach, like the baselines, if downstream training is the goal.
>
> A few more comments about our paper and related work.
>  1. Our model is unique to most of the popular sentence embeddings models as our core contribution is proposing a new generative model. Most prior work uses NLI data or encoder-decoder models to predict different phenomena (i.e. a translation, the next sentence, etc.).
>  2. Much of this related work is trained on substantially more data than our model, or, in the case of Infersent/Sentence-Bert, use word embeddings/BERT which are trained massive data. In contrast, we used just 1M samples from OpenSubtitles/Gigaword which represents a substantially smaller (and noisier) training set -- a more practical use case for most downstream applications. In principle, the use of larger / cleaner data and the incorporation of BERT’s contextualized embeddings is somewhat orthogonal and might stack with our results.
>
> [1] Alex Wang, Amanpreet Singh, Julian Michael, Felix Hill, Omer Levy, and Samuel R. Bowman. GLUE: A multi-task benchmark and analysis platform for natural language understanding. ICLR, 2019.

---

> > ### Author Response · Authors · 2019-11-15
> > **Response to Reviewer #1 (Part 2)**
> >
> > "I’m a bit unclear how these sentence embeddings are translated into a score that decides the degree to which sentences have the same meaning. Is it just cosine similarity of two sentence embedding vectors?"
> >
> > Yes it is just cosine similarity. We did experiment with some others like L2 or L1, but cosine seems to work best (and not just for our models -- for all the models we compare with on STS).
> >
> > "While the purpose of these references is to generate sentences instead of building a sentence embedding, the method is related and comparison and discussion would be worthwhile."
> >
> > Thanks for pointing these out! This was a point also raised by R2. We added experiments (Section 4.4)  with an autoencoder and a monolingual VAE (just using the English side of the en-fr bitext) to help better understand how much performance is due to training on bitext (they are about 7+ points behind BGT on the STS datasets). We hope that this helps better motivate our approach.

---

### Official Review · AnonReviewer3 · 2019-10-29
**Official Blind Review #3**

**Rating:** 3

**Review:**

The paper presents a model that, given parallel bilingual data, separates the common semantics from the language-specific semantics on a sentence level.

Overall the presentation is clear and the experiments show gains over the baselines. One major point of confusion however is that, while early on in the paper (introduction), it is stated repeatedly that one of the strengths of the proposed model is that it is sensitive to word order on a sentence level, this particular aspect of the model is neither evaluated nor analysed. Instead the empirical analysis focuses on sentence length, punctuation and semantics. The analysis of all these three aspects is superficial: for sentence length, it consists of  computing the sentence mean and median; for punctuation,  it consists of masking punctuation; and the last part just computes vectors of nouns only (and states that this is "semantics"). But there is no analysis per se.

Overall, the paper presents a model and shows gains over baselines. The extent to which these gains are due to tuning as opposed to the inherent design of the model is not clear. The analysis is superficial.


**Experience Assessment:**

I have read many papers in this area.

**Review Assessment: Checking Correctness Of Derivations And Theory:**

I did not assess the derivations or theory.

**Review Assessment: Checking Correctness Of Experiments:**

I assessed the sensibility of the experiments.

**Review Assessment: Thoroughness In Paper Reading:**

I made a quick assessment of this paper.

---

> ### Author Response · Authors · 2019-11-15
> **Response to Reviewer #3**
>
> Thank you for the comments! We hope we addressed your concerns below.
>
> “it is stated repeatedly that one of the strengths of the proposed model is that it is sensitive to word order on a sentence level, this particular aspect of the model is neither evaluated nor analysed”
>
> The motivation for these claims was in our performance on the STS Hard datasets (and other data splits like the negation split). These splits were designed to mine sentences from the STS datasets where being able to model ordered word interactions is necessary for strong performance. Specifically, we filtered for examples that had either low symmetric word error rates and low similarity scores (i.e. similar structure and word choice, but low scores) or high symmetric word error rates and high similarity scores. This is described in Section 4.3, and Table 2 and Table 3 show results where our models (and their ablations) achieve much higher performance on these splits relative to prior work.
>
> “The analysis of all these three aspects is superficial”
>
> We have added probing experiments to make this analysis more precise (see our comment to all reviewers).
> These experiments helped us better understand what exactly is being learned by the semantic and language-specific variables and how this changes with language choice. This is described in more detail in 5.2. Note that we also added some style transfer experiments that corroborate the probing results and show an interesting application of our model.
>
> “The extent to which these gains are due to tuning as opposed to the inherent design of the model is not clear”
>
> We hope that our new experiments with different ablations (see comment to all reviewers) illustrate that each component of our model design is contributing to the overall performance, and that all of them together produce the best results. Another addition to our revised version is that we also simplified our model -- there is now no pretraining, no freezing of encoders, and we only have a single hyperparameter (the annealing rate - which just linearly increases). We found our model robust to the annealing rate as well, and we are therefore confident that our results are not due to tuning.

---

### Author Response · Authors · 2019-11-15
**Paper Revision**

We would like to thank all the reviewers for their thoughtful and constructive comments. We have posted a revised version of our paper to both address these comments and also to add additional experiments to further analyze and evaluate our models. Here are some of the highlights:

1. (Section 2,3) We have simplified our model reducing it to a single semantic encoder that is shared between the languages We removed all pretraining as well (as we found that it only led to marginal gains) and now our best model trains from scratch with only a single hyperparameter: the annealing rate of the KL term. Our results are also improved over our original version.
2. (Section 4.4) We added ablations, EnglishTrans, BilingualTrans, BGT w/o LangVars, and BGT w/o Prior to our paper to better understand how each of our design choices cause further improvement over the EnglishTrans baseline. More specifically, BilingualTrans vs. EnglishTrans shows the effect of bilingual training, BGT w/o Prior vs BilingualTrans shows the effect of the KL term regularizing the embeddings space, and BGT w/o  LangVars vs BilingualTrans shows the effect of adding language-specific encoders. When compared to BGT, BGT w/o LangVars and BGT w/o Prior show the effect of having both language-specific variables and priors over the embedding space.
3. (Section 4.4) We added an English-English autoencoder and VAE using our dataset as recommended by R1 and R2. These experiments show the contribution that parallel text gives and better motivates our model.
4. (Section 5.2) We added further analysis through probing experiments to better quantify the behavior of  the semantic and language-specific encoders. These probing tasks cover semantic, syntactic properties in addition to length, word content, punctuation, and classifying correct/incorrect gender (in Romance languages). These results corroborate the analysis in the initial version of the paper, and we hope that you find them more convincing.
5. (Section 5.2) We also trained bilingual models in other languages (Spanish, Arabic, Turkish, and Japanese) and found consistently better results with our model over strong baselines. We also analyze the effect of language choice on information encoded by the semantic and language-specific encoders, showing that it can have a pronounced impact.
6. (Section 5.3) Lastly, we also added some qualitative examples of language generation from our generative model. These results give another view of the information captured by the semantic and language-specific variables. We feed different sentences in each encoder (semantic and language-specific), and then see how the output relates to the source sentences. The results indicate that content is encoded by the semantic variable, while roughly “stylistic” attributes are encoded by the language-specific variable -- drawing a connection with results from style transfer.

---

### Decision · Program_Chairs · 2019-12-19

**Decision:**

Reject

**Comment:**

This paper presents a model for building sentence embeddings using a generative transformer model that encoders separately semantic aspects (that are common across languages)  and language-specific aspects. The authors evaluate their embeddings in a non-parametric way (i.e., on STS tasks by measuring cosine similarity) and find their method to outperform other sentence embeddings methods. The main concern that both reviewers (and myself) have about this work relates to its evaluation part. While the authors present a set of very interesting difficult evaluation and probing splits aiming at quantifying the linguistic behaviour of their model, it is unsatisfying the fact that the authors do not evaluate their model extensively in standard classification embedding benchmarks (e.g., as in GLUE). The authors comment: “[their model in producing embeddings] it isn’t as strong when using classification for final predictions. This indicates that the embeddings learned by our approach may be most useful when no downstream training is possible”. If this is true, why is it the case and isn’t it quite restrictive? I think this work is interesting with a nice analysis but the current empirical results are borderline  (yes, the model is better on STS, but this is quite limited of an idea compared to using these embeddings as features in a classification tasks). As such, I do not recommend this paper for acceptance but I do hope that authors will keep improving their method and will make it work in more general problems involving classification tasks.